# Vision and Vibration Data Fusion-Based Structural Dynamic Displacement Measurement with Test Validation

**DOI:** 10.3390/s23094547

**Published:** 2023-05-07

**Authors:** Cheng Xiu, Yufeng Weng, Weixing Shi

**Affiliations:** Department of Disaster Mitigation for Structures, College of Civil Engineering, Tongji University, Shanghai 200092, China; xiucheng@tongji.edu.cn (C.X.);

**Keywords:** Kanade–Lucas–Tomasi optical-flow method, data fusion, computer vision, Kalman filter

## Abstract

The dynamic measurement and identification of structural deformation are essential for structural health monitoring. Traditional contact-type displacement monitoring inevitably requires the arrangement of measurement points on physical structures and the setting of stable reference systems, which limits the application of dynamic displacement measurement of structures in practice. Computer vision-based structural displacement monitoring has the characteristics of non-contact measurement, simple installation, and relatively low cost. However, the existing displacement identification methods are still influenced by lighting conditions, image resolution, and shooting-rate, which limits engineering applications. This paper presents a data fusion method for contact acceleration monitoring and non-contact displacement recognition, utilizing the high dynamic sampling rate of traditional contact acceleration sensors. It establishes and validates an accurate estimation method for dynamic deformation states. The structural displacement is obtained by combining an improved KLT algorithm and asynchronous multi-rate Kalman filtering. The results show that the presented method can help improve the displacement sampling rate and collect high-frequency vibration information compared with only the vision measurement technique. The normalized root mean square error is less than 2% for the proposed method.

## 1. Introduction

In structural health monitoring, it is necessary to deploy sensors to monitor the structure’s response. These sensors collect important data on various aspects of the structure, such as vibrations and displacements [1,2]. These measurements can provide valuable insights into the structure’s integrity and indicate any load anomalies or structural defects. Moreover, displacement monitoring can also be used to update the finite element model of the structure, which is essential for accurately assessing, monitoring, and controlling civil infrastructure [3,4,5,6,7]. For example, peak deformation demands, including peak inter-story drift ratio and peak roof displacement, are essential indicators in earthquake engineering for evaluating structural seismic performance [8,9,10,11]. Vehicle-induced displacement is also utilized to detect bridge damage and assess bridge conditions [12]. Additionally, the displacement of a high-rise building is an important indicator of safety [13,14]. Therefore, displacement is critical in ensuring civil infrastructure’s health and integrity.

There are many means of directly measuring the displacement response of a structure in the field of structural engineering, which include pull-wire displacement gauges, linear variable differential transformers (LVDT) [15], laser Doppler vibrometers (LDV) [16], Real-Time Kinematic global satellite navigation systems (RTK-GNSS) [17], etc. LVDT usually need to be installed between the target point and a fixed reference point; hence, despite the high accuracy of LVDT measurements, they are not easy to be installed in practical engineering [18,19]. As LVDT is a contact measurement method, any severe structural deformation or breakage during a shaking table test can potentially damage the LVDT. On the other hand, LDV can remotely measure displacement with high resolution and accuracy; however, it can be expensive and limited to a few measurement points [16]. RTK-GNSS is more accurate than normal GNSS and can provide centimeter-level accuracy; however, it has a lower sampling frequency [20]. Moreover, the GPS method is infeasible for indoor measurement due to the requirement of signal reception [21,22]. Accelerometric integration is also used to measure displacement; however, this method suffers from low-frequency drift and cannot measure residual deformation [23]. Several methods [24,25,26,27,28] have been used to solve the drift problem; however, these methods will remove information about the structure’s response.

With the development of high-quality, low-cost optical cameras and lenses in recent years, structural monitoring and inspection–based computer vision have gradually become a hot topic. Numerous displacement estimation methods, such as template matching [29,30,31], feature matching [18,32,33,34], digital image correlation (DIC) [35,36,37], and optical flow methods [38,39,40,41], have been proposed. Optical flow methods are widely used among these techniques due to their high accuracy and computational efficiency. Many researchers have utilized the Kanade–Lucas–Tomasi (KLT) tracker, an intensity-based optical flow estimation algorithm, for target-based or target-free structural displacement measurement [42,43,44]. The concept of optical flow was initially introduced by Gibson [45] and referred to the velocity of a moving object in a time-varying image. Based on this idea, the KLT optical flow method matches and tracks feature points in two adjacent frames to obtain motion information for those points. However, despite its advantages, the KLT method has two primary limitations, loss of feature points during tracking [41,46] and drift-type errors [47,48], respectively. Regarding the first limitation, since the Taylor expansion is used in the derivation of KLT, it is necessary to satisfy the assumption of small deformations, which is described in detail in Section 2. As for the second limitation, the KLT tracker estimates the feature locations by using an image gradient, but errors induced by integration drift can cause inaccuracies in the measurement of residual displacement, leading to deviations from the correct tracking over time. This problem is particularly challenging when tracking long sequences, even though it may not be noticeable in individual image pairs.

To estimate displacement in a high sampling rate, vision-based measurements at a low sampling rate and acceleration measurements at a high sampling rate can be combined as well. The study by Roberts et al. [49] highlighted the importance of displacement fusion in extending the available frequency band, particularly in detecting vibrations of bridges reliably. They found that a minimum sampling rate of 100 Hz is required for bridges. To achieve the sampling rate, several researchers have proposed to fuse low-sampled measurements, such as GPS and strain sensors, with high-sampled measurements (such as acceleration) [50,51,52,53]. Efforts were also taken to fuse vision cameras and accelerometers. Park et al. [54] utilized a complementary filter to fuse acceleration and displacement, while Ma et al. [55] employed an adaptive Kalman filter to estimate displacement. These methods mainly used the feature-matching based method to estimate displacement, which takes more time compared with the KLT method.

Utilizing the high dynamic sampling rate of traditional contact acceleration sensors, this paper introduces a data fusion approach for contact acceleration monitoring and non-contact displacement recognition, constructing and validating an accurate estimation method for critical dynamic deformation states in structures. This paper is structured as follows: Section 2 provides a brief overview of the KLT algorithm. Section 3 introduces the algorithm employed in this study. Section 4 presents the study’s results, demonstrating the proposed method’s high efficiency, accuracy, and robustness in achieving drift-free large structural displacements. The primary limitations of the KLT method were addressed by fusing accelerometer data, which improved the accuracy of feature tracking and reduced errors caused by integration drift. Finally, the concluding remarks are presented in Section 5.

## 2. A Brief Review of the Kanade–Lucas–Tomasi (KLT) Method

Optical flow refers to the pattern of apparent motion of objects in an image between two frames due to either the motion of the object or the camera. For instance, in Figure 1, three target points in two adjacent images can have their positions in the second image identified by detecting the pixels with consistent intensity values with the corresponding pixels in the first image. It represents the displacement of a 2D vector field $\left(d_{x},d_{y}\right)$ when a feature point moves from the first frame I(x,y,t) to the second frame after a time interval of $d_{t}$. The optical flow equation assumes that the object’s brightness does not change.
(1)$$I_{1}(x,y,t)=I_{2}(x+d_{x},y+d_{y},t+d_{t})$$

$I_{1}(x,y,t)$ represents image pixels from the reference image, and $I_{2}(x+d_{x},y{+d}_{y},t+d_{t})$ is the image pixels of the following image. For simplicity, let $d={\left[d_{x},d_{y}\right]}^{T}$, $X={\left[x,y\right]}^{T}$.

**Figure 1 sensors-23-04547-f001:**
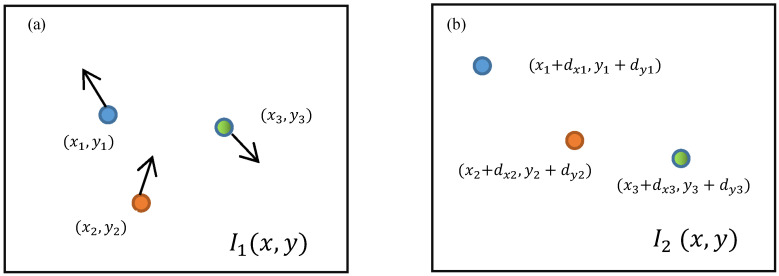
Demonstration of motion of three points in two frames: (**a**) first frame and (**b**) second frame.

Under the pixel window, the error function is constructed:(2)$$\varepsilon =\iint_{W}{\left[I_{2}(X+d)-I_{1}(X)\right]}^{2}\omega (X)dX$$
where a window W, centered on the position of a target point, is established in the first image. w(X) is a weighting function that assigns weight to the surrounding pixel. In the simplest scenario, w(X)=1. Another commonly used function is the Gaussian function, which addresses the center of the window.

Set the partial derivative of ε with respect to d as:(3)$$\iint_{W}\left[I_{2}\left(X+d\right)-I_{1}\left(X\right)\right]\left[\frac{\partial I_{2}\left(X+d\right)}{\partial d}-\frac{\partial I_{1}\left(X\right)}{\partial d}\right]\omega (X)dX=0$$

The following formula can be obtained from Taylor’s expansion:(4)$$I_{2}\left(X+d\right)\approx I_{2}(X)+d_{x}\frac{\partial I_{2}}{\partial x}(X)+d_{y}\frac{\partial I_{2}}{\partial y}(X)$$

The substitution of Equation (4) into Equation (3) leads to
(5)$$\iint_{W}\left[I_{2}(X)-I_{1}(X)+p^{T}d\right]p(X)\omega (X)dX=0$$
where:(6)$$p={\left[\frac{\partial I_{2}}{\partial x},\frac{\partial I_{2}}{\partial y}\right]}^{T}$$

The following equation can be obtained from Equation (5):(7)Zd=e
where $Z=\iint_{W}p(X)p^{T}(X)\omega (X)dX$**,** and $e=\iint_{W}\left[I_{1}(X)-I_{2}(X)\right]p(X)\omega (X)dX$.

Equation (7) is solved by an iterative method to obtain the value of d. When the value of e is less than the set threshold, the approximate solution of d can be obtained. In summary, the KLT tracker uses points from the previous and current frames to create motion vectors. Selecting these feature points is an essential part of the KLT method. Normally, a region-of-interest (ROI) is used to focus on a specific part of an image to extract relevant information. Common feature detectors include scale-invariant feature transform (SIFT), speeded-up robust features (SURF), and oriented FAST and rotated BRIEF (ORB) [56]. The Harris point suggested in [44] is an efficient detector in real-time for calculating the optical flow because Harris points are simple, reliable, and efficient corner detection. Traditionally, the KLT algorithm calculates velocity by computing the optical flow between consecutive frames. If the small motion assumption is not satisfied, the traditional way is using an image pyramid, as shown in Figure 2, which is briefly described below.

The overall pyramidal tracking algorithm proceeds as follows: as an initial layer 0, the original image is used, and the image is reduced by $2^{L}$ times in length and width to serve as a layer L. The Gaussian pyramid is generated using the obtained images by superimposing them from bottom to top. The corresponding points are also reduced by $2^{L}$ times. The displacement value of the target point on the highest layer is calculated using the method described in the previous section. This value is used in the optical-flow calculation of the next layer as an initial guess to determine the accurate displacement value. Once the displacement value is calculated, it is passed to the following layer as an initial guess and then to the lowest layer (level 0) to obtain the actual displacement value. The work of Kim et al. [57] provides a detailed description of the propaganda process. The limitations of the KLT method are discussed and demonstrated by Won et al. [41] The paper demonstrates feature loss and drift occurrence in the KLT method.

## 3. Methodology

Figure 3 shows an overview of the proposed method. As presented in Figure 3a, one camera is fixed on the ground to trace natural targets on the structure, and an accelerometer is placed on the same floor as the natural targets. Figure 3b illustrates the two stages of the proposed technique for displacement estimation. In the first stage, referred to as the calibration stage, shown in Figure 3, several tasks are accomplished, including the correction of lens parameters, time synchronization, and scale factor calculation. Following this, the second stage, which is called the displacement estimation stage, is initiated.

### 3.1. Calibration Stage

#### 3.1.1. Video Preprocessing and Measurement Conversion

This section uses video preprocessing to correct the distortion caused by the wide-angle lens typically used in consumer-grade cameras. A chessboard pattern is used to calibrate the camera to correct lens distortion [58]. The calibration process involves capturing multiple chessboard images from different angles and orientations, enabling the estimation of the parameters for the lens distortion model. Once the distortion parameters are determined, the images are rectified to remove the distortion and create a rectified image. 

#### 3.1.2. Time Synchronization between Vision and Acceleration

This study used two separate acquisition systems to collect data from the camera and the accelerometer.Due to varying sampling rates and data sources, time synchronization is critical before fusing them. As a result, it was necessary to synchronize the data in time. As shown in Figure 4, to avoid the low-frequency drift phenomenon commonly observed in acceleration sensors, the integration results were filtered using a bandpass filter. The lower limit of the passband in bandpass filtering should be sufficiently large to avoid drift, and the upper limit should be at 1/10 of the camera sampling frequency [59]. Additionally, the results of computer vision measurements were resampled to match the sampling frequency of the acceleration measurements. The computer vision measurement results were also filtered using a bandpass filter with the same range as the integration results. This step reduces the impact of frequencies outside the filter range. The cross-correlation analysis was used to finely align the data from the camera and the accelerometer [54]. Here, the time lag is determined at the point where the maximum value of the cross-correlation occurs. This process enabled accurate data matching from both systems and properly synchronized the recorded data.

#### 3.1.3. Calculating the Scale Factor

The scale factor λ, determined by the distance between the camera and the target object, translates the image pixel values into real-world metric values, as shown below.
(8)$$\lambda =\frac{D}{d}(\mathrm{unit}:\frac{\mathrm{mm}}{\mathrm{pixel}})$$
where D is the actual dimension of the known object, and d is the number of pixels in the image that covers the object.

After time synchronization, the displacements obtained from both methods are truncated to the same length. The scale factor is then estimated using the least squares method. By implementing these steps, potential discrepancies in the displacements can be minimized, and the study results can be reliable.

### 3.2. Displacement Estimation Stage

#### 3.2.1. Drift-Free KLT Method

Figure 5 describes the detailed procedure for estimating target displacement in the i-th frame. It is important to note that the proposed technique only applies to in-plane motion estimation, and only one direction is considered, though it can be extended in two directions. The method includes the following steps: first, feature points, such as Harris corner points, are selected in the reference frame. Using a priori estimate y, the current frame image is translated. Image translation allows for the adjustment of the images in a way that the displacements fall within the range of small motion, enabling the application of the Taylor expansion of Equation (4).

Consequently, this approach improves the accuracy and reliability of the displacement estimation, particularly in cases where the initial displacements may not meet the small motion assumption. Furthermore, by incorporating image translation, the proposed method demonstrates its adaptability to various scenarios, enhancing its practical applicability and performance. After translating the image, the KLT algorithm calculates the optical flow between the reference frame and the current frame to obtain the average velocity of the selected feature points, which is used to determine their average displacement.
(9)$$d=d_{translate}+d_{KLT}$$

In the Equation (9), d is the displacement of different frames, and $d_{KLT}$ is displacement calculated from the drift-free KLT method. $d_{translate}$ is the image translate pixel, calculated as follows:(10)$$d_{translate}=round(\frac{D_{predicted}}{\lambda })$$
where $D_{predicted}$ is the predicted displacement of the target object. Using a priori estimation in the proposed method improves the accuracy of displacement estimates by minimizing the impact of drift-type errors that can accumulate over time. Furthermore, by selecting feature points with strong texture in the reference frame and employing optical flow to calculate displacement, the method further improves the accuracy of displacement estimates.

#### 3.2.2. Asynchronous Kalman Filter

The Kalman filter is a widely used method for data processing that estimates data by continuously predicting and correcting in the time domain. In general, the sampling frequency of the accelerometer is higher than the frame rate of the video. Smyth and Wu [60] used a multi-rate Kalman filter to fuse acceleration and displacement at different sampling rates to improve the estimation of the displacement signal. Ma et al. [55] proposed an asynchronous Kalman filer to fuse acceleration and displacement with adaptive parameters.

In the case of asynchronous situations, Ma et al. [55] categorized time steps into three types. Figure 6 shows the overview of the proposed methods. Type 1 involves only acceleration updates, while the second type involves visual updates. Type 3 involves acceleration updates following visual updates. Among these three types, only in type 2 are the values and probabilities of displacement fused when computing computer displacement updates.

Suppose $X_{k}=[x_{k},{\dot{x}}_{k}{]}^{T}$ is a state variable, and $x_{k},{\dot{x}}_{k}$ represents displacement and velocity, respectively, at the k-th time step, then a discrete state space model for the relationship between acceleration and displacement can be described as:(11)$$X_{k}=A\left(dt\right)X_{k-1}+B\left(dt\right)a_{k-1}+B\left(dt\right)w_{k-1}$$
(12)$$D_{k}=HX_{k}+v_{k}$$
where $w_{k}$ and $v_{k}$ are the noises of measured acceleration and displacement, respectively. Q and R are the corresponding variances of $w_{k}$ and $v_{k}$, respectively. dt is the time interval of the time step. A and B are the state transition matrix and control input matrix, respectively. In this case, they are functions of the time interval:(13)$$A\left(dt\right)=\left[\begin{array}{cc} 1 & dt\\ 0 & 1 \end{array}\right];B\left(dt\right)=\left[\begin{array}{c} {dt}^{2}/2\\ dt \end{array}\right];H=\begin{array}{cc} {}[1 & 0 \end{array}]$$

Assume that during type 1, only acceleration is considered. The ${\hat{X}}_{k}^{-}$ and its covariance ${\hat{P}}_{k}^{-}$ were obtained as follows:(14)$${\hat{X}}_{k}^{-}=A\left(dt_{a}\right){\hat{X}}_{k-1}^{+}+B\left(dt_{a}\right)a_{k-1}$$
(15)$${\hat{P}}_{k}^{-}=A\left(dt_{a}\right){\hat{P}}_{k-1}^{-}A^{T}\left(dt_{a}\right)+Q\left(dt_{a}\right)$$
(16)$$Q\left(dt_{a}\right)=q\left[\begin{array}{cc} {dt}_{a}^{3}/3 & dt_{a}^{2}/2\\ dt_{a}^{2}/2 & dt_{a} \end{array}\right]$$
where $dt_{a}$ and q denote the time interval and noise variance of the acceleration measurements, respectively. The q value can be easily estimated using laboratory testing.

Since no other measurement is available in this time interval,
(17)$${\hat{X}}_{k}^{+}={\hat{X}}_{k}^{-};{\hat{P}}_{k}^{+}={\hat{P}}_{k}^{-}$$

In type 2, the prior state ${\hat{Y}}_{i}^{-}$ and covariance ${\hat{G}}_{i}^{-}$ can be estimated according to the following state estimation:(18)$${\hat{Y}}_{i}^{-}=A\left(dt_{k,i}\right){\hat{x}}_{k}+B\left(dt_{k,i}\right)a_{k}$$
(19)$${\hat{G}}_{i}^{-}=A\left(dt_{k,i}\right){\hat{P}}_{k}A^{T}\left(dt_{k,i}\right)+Q(dt_{k,i})$$
where $dt_{k,i}$ denotes the time interval between the k-th acceleration and the i-th vision measurements. With the ${\hat{Y}}_{i}^{-}$, the drift-free KLT method was applied to estimate displacement $d_{i}$ from vision measurements.

The posterior state and its covariance were calculated as follows:(20)$${\hat{Y}}_{i}^{+}={\hat{Y}}_{i}^{-}+{\hat{P}}_{k}H^{T}{\left(H{\hat{P}}_{k}H^{T}+R\right)}^{-1}(D_{i}-H{\hat{Y}}_{i}^{-})$$
(21)$${\hat{G}}_{i}^{+}=(I-{\hat{P}}_{k}H^{T}{\left(H{\hat{P}}_{k}H^{T}+R\right)}^{-1}){\hat{G}}_{i}^{-}$$

Here, R is calculated as follows:(22)$$R=\sigma_{D}^{2}/dt_{k,i}$$
where $\sigma_{D}^{2}$ is the observation noise of displacement measurement.

In type 3, the prior state and covariance are estimated according to the following state estimation:(23)$${\hat{X}}_{k+1}=A\left(dt_{i,k+1}\right){\hat{Y}}_{i}^{+}+B\left(dt_{i,k+1}\right)a_{k}$$
(24)$${\hat{P}}_{k+1}=A\left(dt_{i,k+1}\right){\hat{G}}_{i}^{+}A^{T}\left(dt_{i,k+1}\right)+Q\left(dt_{i,k+1}\right)$$

#### 3.2.3. Parameter Estimation

As described in Equation (8), the actual displacement is the product of the scale factor and the pixel displacement. Therefore, according to the law of error transfer, the variance of the displacement measurement can be calculated by the following equation:(25)$$\sigma_{D}^{2}=\sigma_{\lambda }^{2}\cdot {\bar{D}}^{2}+\sigma_{D}^{2}\cdot {\bar{\lambda }}^{2}$$
where $\sigma_{u}^{2}$ is the variance of the displacement measurement, $\bar{D}$ and $\sigma_{D}^{2}$ are the mean and variance of the displacement, respectively, and $\bar{\lambda }$ and $\sigma_{\lambda }^{2}$ are the mean and variance of the scale factor, respectively. For structural monitoring, the mean value of displacement $\bar{d}$ can be assumed to be 0, and the variance of displacement $\sigma_{d}^{2}$ is estimated by calculating the mean of the variance of all frames based on the matching results for each frame.

### 3.3. Comparison with Conventional Motion Estimation Approaches

This paper compares two commonly used motion estimation methods: (a) a feature-matching-based method [32] and (b) the commonly used KLT tracker, as mentioned in Section 2. The feature-matching-based method consists of the following steps: (1) video preprocessing: the specific step is the same as the method in this article. (2) Feature detection and feature description: this step detects distinctive features or key points in ROI. These features are usually corners, edges, or regions with rich textures. Here, the Harris corner detection algorithm is used. After detecting the features, a descriptor is computed for each feature. (3) Feature matching: the matching step involves comparing the descriptors from the two images and finding the best match for each feature. (4) Outlier removal: since not all matched features correspond to the same physical point in the scene, some matches might be incorrect or outliers. One effective technique for outlier removal is random sample consensus (RANSAC), which is commonly used in computer vision and image processing. (5) Motion estimation: the relative displacement between the two images can be computed with the set of correctly matched features.

## 4. Small-Scale Laboratory Validation

### 4.1. Experimental Setup

The proposed method for drift-free large motion measurement is investigated in a laboratory experiment to determine its performance and its sensitivity to the video’s frame rate. Figure 7 illustrates the validation of a three-story steel building model excited by a uniaxial shaking table. The simultaneous measurement of structural responses was conducted using the proposed system and a laser displacement sensor used for ground truthing; the details regarding these devices are in Table 1. The algorithm was implemented in MATLAB, running on a PC with a 2.3 GHz Intel i7 processor and 32 GB of RAM. In this experiment, three different types of excitations were used at the bottom of the structure: (1) 1 Hz sine excitation, (2) 4 Hz sine excitation, and (3) earthquake excitation.

### 4.2. Experimental Result

To quantify the measurement accuracy of the results, the error analysis is conducted using the normalized root-mean-square error (*NRMSE*):(26)$$NRMSE=\frac{\sqrt{\frac{1}{N}\sum_{i=1}^{N}{\left({\hat{x}}_{i}-x_{i}\right)}^{2}}}{max(x_{i})-min(x_{i})}$$
where $\hat{x}$ is the estimated displacement; x is the reference displacement; N is the number of displacement measurements.

Figure 8 shows the grayscale initial video frame; the selected target region is framed in a red box containing the salient corner features to be tracked. This figure shows that the Harris detector successfully detects the corner of the structure and other feature points. After selecting the feature points, the feature-matching-based method is employed to estimate the movement of the target object for each frame of the video in the calibration stage.

Under case (1), the scale factor calculation results proposed in this paper are shown in Figure 9, while the scale factor obtained through structural size measurement is 0.78 mm/pixel. The two-scale factor shows that the scale method estimated here is effective, so in the absence dimension scenario, the scale factor can be estimated in this way.

In the Kalman filter process, the noise parameter q is selected as $10^{4}\;{mm}^{2}/s^{2}$ in this experiment, which is estimated based on prior experience. For case (1), with a video sampling rate of 100 Hz, the results are shown in Figure 10. As shown in the figure, compared to the feature-based and KLT methods, the KLT method exhibits a significant drift phenomenon, while the drift-free method proposed in this paper does not have this issue. All comparisons are made here by linearly interpolating the data to 500 Hz. The method proposed in this paper can improve the NRMSE value, reducing it by 38% and 83%, respectively.

Figure 11 shows that the target is within ROI using the proposed image translating method. In this figure, the frame below moves 16 pixels, and the target in the ROI roughly remains the same. Thus, the effectiveness of image translation during significant displacement is verified.

The vision sample frequency was modified to investigate the influence of the vision sample frequency further and reduce computation time. In case (1), by resampling the video and reducing the sampling frequency to 50 Hz, 25 Hz, and 10 Hz, it can be found that as the sampling frequency decreases, the NRMSE values increase to 0.91%, 1.52%, and 1.51%, respectively, as shown in Figure 12. In comparison, the feature-matching method changes to 1.57%, 1.66%, and 2.51%, respectively, and the result for the KLT method is higher than the feature-matching method. In case (1), since the excitation frequency is only 1 Hz, the forced vibration frequency can be accurately captured in all cases. 

In case (2), the input frequency at the base of the structure was set to 4 Hz, with an amplitude of 30 mm, allowing for the evaluation of the effectiveness of the proposed method under large displacement and high-frequency vibration conditions. As in previous tests, the laser displacement sensor at the top of the structure was used as a reference for calculating the error values. This experimental setup aimed to demonstrate the accuracy and reliability of the proposed method in large amplitude and high-frequency vibrations.

Figure 13 shows the displacement of 100 Hz and 10 Hz sample frequencies. As the figure shows, compared with 100 Hz, the pure 10 Hz vision sample frequency failed to capture several peak values. Under a frequency of 100 Hz, the NRMSE values for the proposed method and the feature matching method were 1.3% and 1.58%, respectively. At a sampling frequency of 10 Hz, the KLT method could not detect displacements and was, thus, omitted from the comparison. The NRMSE values for the proposed and feature matching methods at 10 Hz were 5% and 12%, respectively. These findings indicate that for high-frequency vibrations, the accuracy of purely visual methods is limited due to the constraints imposed by the Nyquist sampling theorem, preventing real-time data acquisition.

The computational time for each frame is presented in Table 2. Table 2 reveals that the proposed method’s computation time is shorter than the feature-matching method but longer than the KLT algorithm. In principle, the computation time for the proposed method should be close to that of the KLT algorithm. The discrepancy in computation times may be attributed to the time required for image translation and algorithm initialization. Further investigation into optimizing the proposed method’s computation time may help close the gap and make it more comparable to the KLT algorithm, enhancing its practical applicability in real scenarios. Additionally, the proposed method can provide real-time estimations of drift-free displacements due to the reduced computation time. This advantage makes the method more suitable for applications where rapid and accurate displacement measurements are critical. The proposed method can outperform alternative approaches, particularly in scenarios with high-frequency vibrations or large displacements, by offering a balance between accuracy and efficiency.

Case (3) presents the displacement of the frame under the excitation of the El Centro earthquake wave. Due to the frame’s flexibility, unlike case (1) and case (2), the top-floor displacement is primarily governed by the frequency of the structure. In Case (3), the time history curves and NRMSE values were calculated for different video sampling rates, as shown in Figure 14. As the sampling rate decreases, the NRMSE increases from 0.83% to 0.93%, 0.91%, and 1.13%. This trend demonstrates the influence of sampling rate on the accuracy of displacement measurements.

Under earthquake conditions, the NRMSE values decrease for the given conditions, indicating that the proposed method exhibits robustness. This improved performance demonstrates the method’s ability to maintain accuracy and reliability even in challenging situations. The method’s robustness is crucial in practical applications, where dynamic conditions and external disturbances can significantly impact the quality and reliability of displacement estimates.

Power spectral density (PSD) is a function used to describe the energy distribution of a signal in the frequency domain. PSD is frequently used in signal processing and communication systems to describe the spectral characteristics of noise and signals.

Figure 15 shows that the proposed method’s PSD is closer to the reference measurement results. Note that here the PSD is calculated without interpretation. The high-frequency information of the structure is more similar to the LDV results, which is beneficial for determining the structure’s frequency and mode shapes. For the low-frequency portion, after applying the Kalman filter, the power spectral density curve is closer to the pure visual results. The first frequency, 2.63 Hz, is successfully identified in both scenarios. Under the 10 Hz scenario, the vision method failed to identify the 6.84 Hz, the second mode. The third frequency, 18.55 Hz, is not apparent from those curves. These results demonstrate the effectiveness of incorporating the Kalman filter in improving the accuracy of the displacement estimates, particularly in capturing the structure’s essential dynamic characteristics across various frequency ranges.

## 5. Conclusions

This paper uses an accelerometer and computer vision techniques to fuse contact monitoring and non-contact tracking data of structural dynamics to exploit both advantages fully. In response to the shortcomings that computer vision techniques cannot capture high-frequency vibration information of structures and require additional parameters to estimate the scaling factor, while accelerometers cannot monitor low-frequency displacements and have zero drift, this paper proposes to fuse data from computer vision and accelerometers using Kalman filtering to calculate the scaling factor using the least squares method.

The method’s reliability is also verified by using a frame structure shaker table test. The results show that (1) the method can reliably estimate the scale factor. (2) In the time domain, the NRMSE value is effectively reduced, and the overall displacement measurement accuracy is improved. (3) In the frequency domain range, the proposed data fusion method compensates for the low sampling rate of pure computer vision and effectively improves the signal-to-noise ratio of displacement data in the higher-order mode range.

The study also investigated the impact of lowering the sampling frequency on the video vision technique. The findings reveal that the accuracy of the displacements is only slightly affected when the sampling frequency is decreased from 100 to 10 Hz. The fused displacements’ power spectral densities remain unchanged, even though the sampling frequency is reduced to a tenth of its original value. This demonstrates that the proposed fused method is a feasible and efficient alternative for measuring displacement in civil engineering structures.

## Figures and Tables

**Figure 2 sensors-23-04547-f002:**
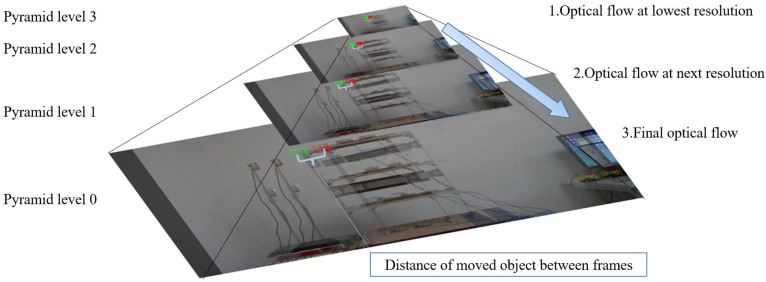
Image pyramid of the KLT method.

**Figure 3 sensors-23-04547-f003:**
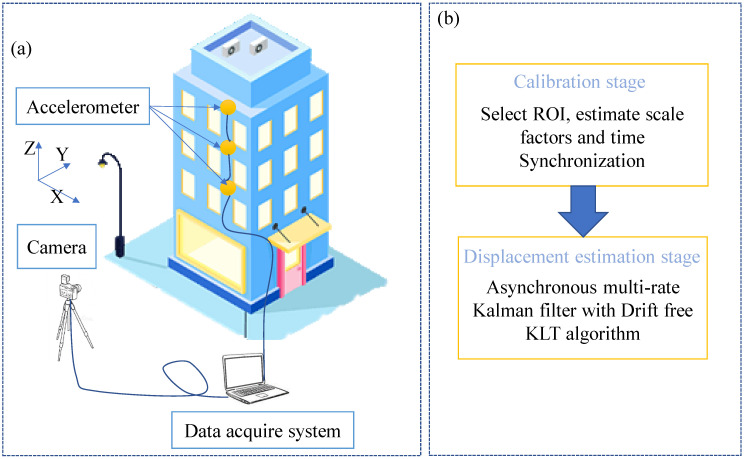
Overall of the proposed displacement estimation technique (**a**) sensors and camera layout (**b**) main stages of the proposed technique.

**Figure 4 sensors-23-04547-f004:**
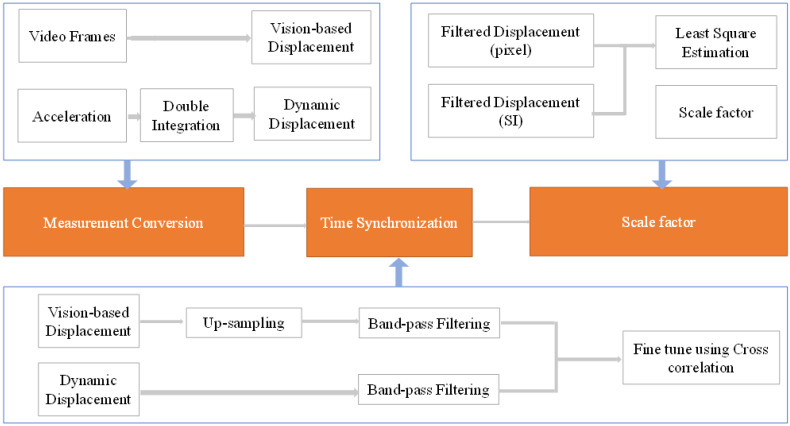
The overall process of time synchronization and calculating the scale factor.

**Figure 5 sensors-23-04547-f005:**
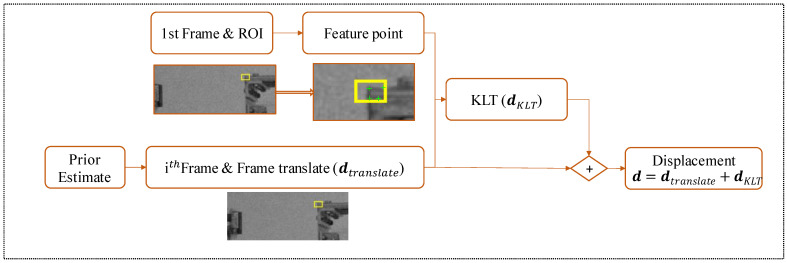
Proposed drift-free KLT method.

**Figure 6 sensors-23-04547-f006:**
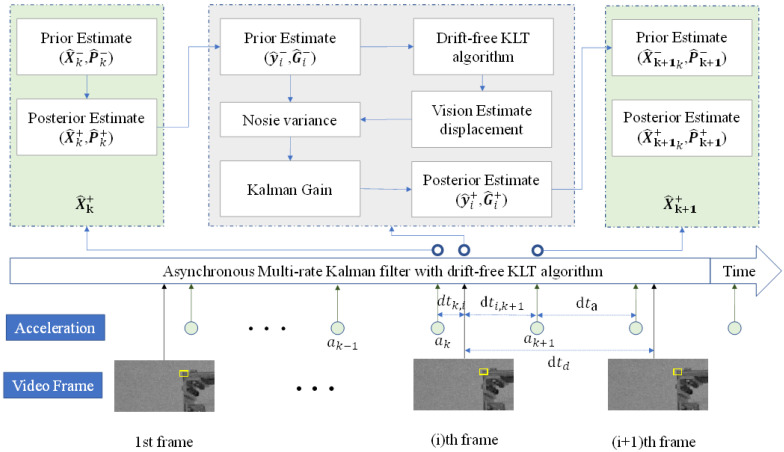
Overview of how to fuse asynchronous vision and acceleration measurement using an asynchronous multi-rate Kalman filter with drift-free KLT algorithm to estimate structural displacement.

**Figure 7 sensors-23-04547-f007:**
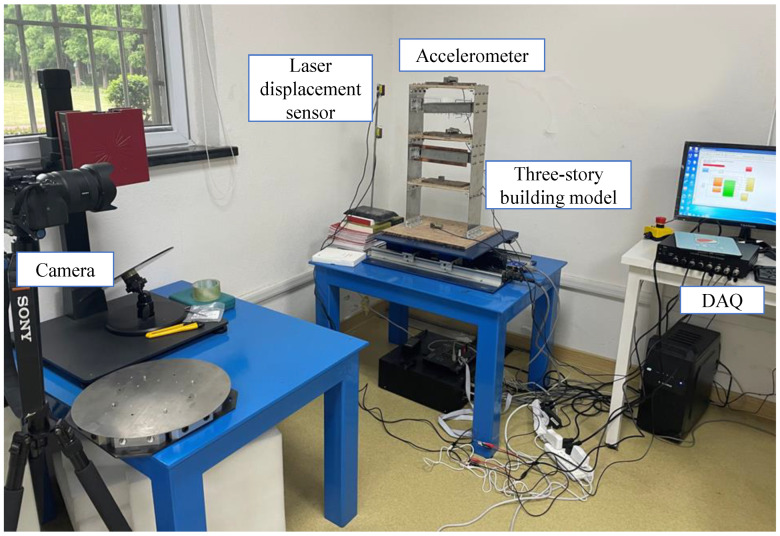
Experimental setup of shaking table test.

**Figure 8 sensors-23-04547-f008:**
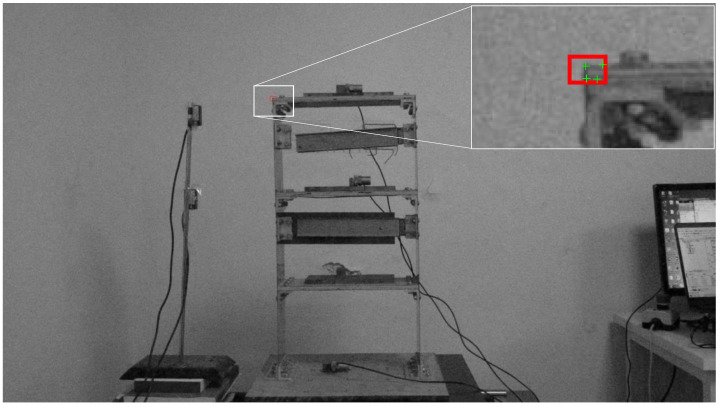
Selected target (red box) at initial video frame and initial track point (green point).

**Figure 9 sensors-23-04547-f009:**
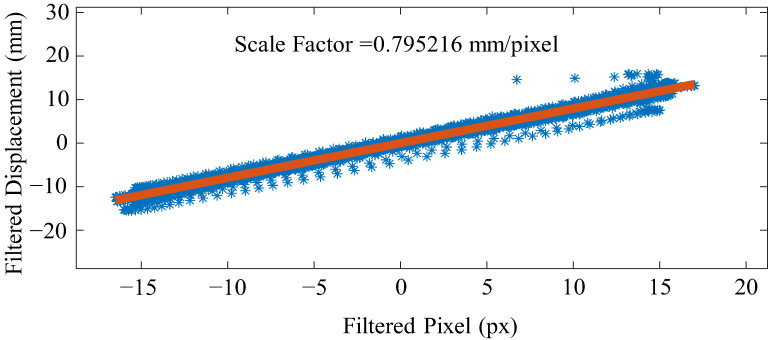
Scale factor of the experiment.

**Figure 10 sensors-23-04547-f010:**
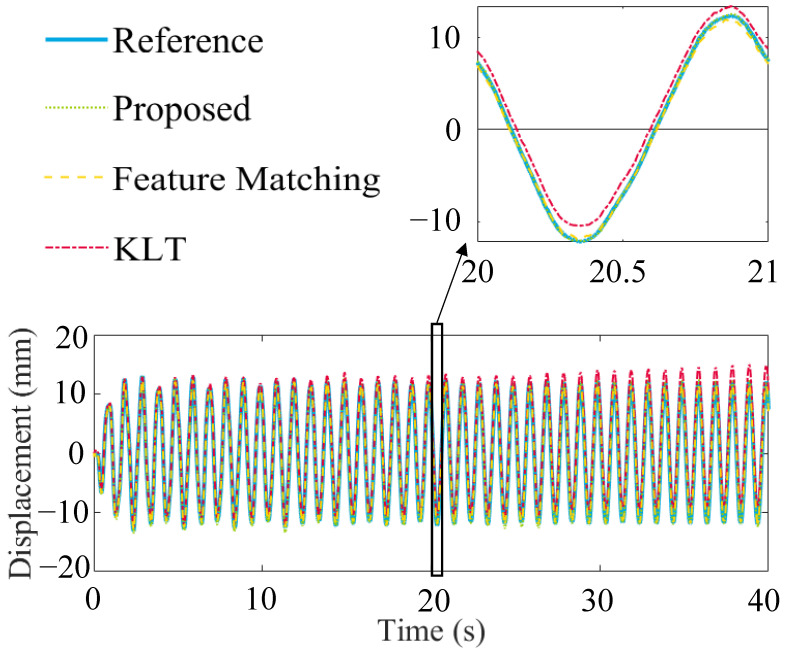
Comparison between ground truth and displacements estimated by vision algorithms.

**Figure 11 sensors-23-04547-f011:**
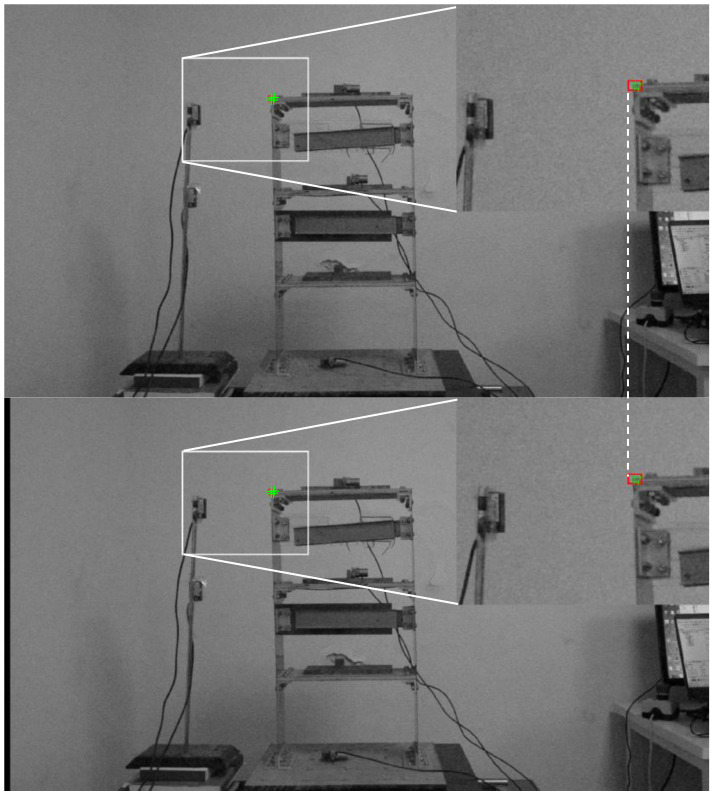
Comparison of ROI at different steps of the case (1). The red box represents the ROI, and the green dots represent the feature points.

**Figure 12 sensors-23-04547-f012:**
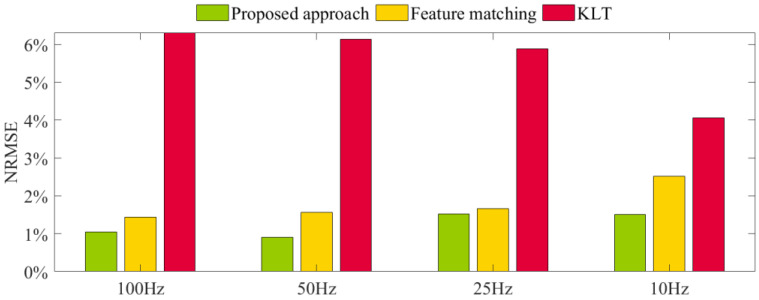
NRMSE of different sample frequency.

**Figure 13 sensors-23-04547-f013:**
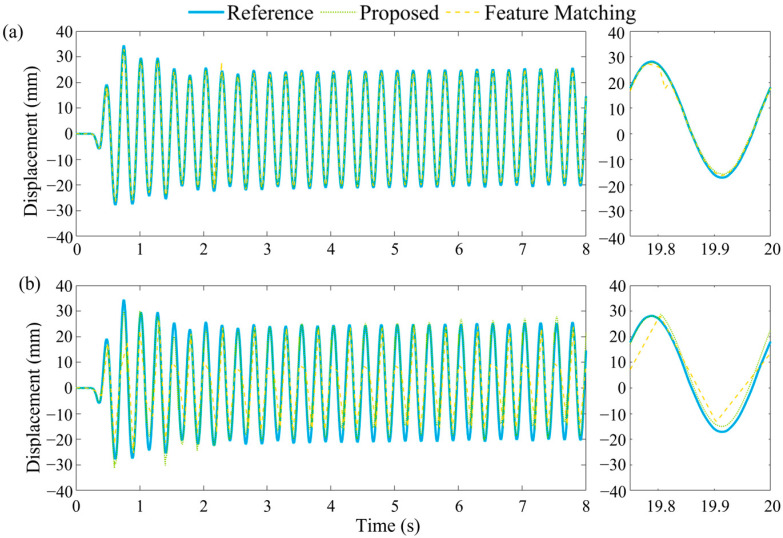
Comparison between ground truth and estimate displacement with different vision sample rates in case (2) (**a**) 100 Hz and (**b**) 10 Hz.

**Figure 14 sensors-23-04547-f014:**
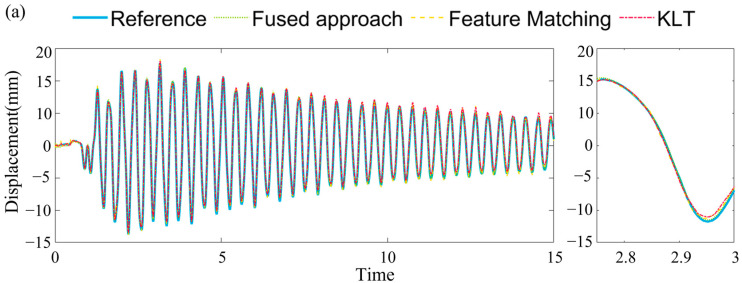
(**a**) time history curve for earthquake excitation (**b**) NRMSE of different sample frequency.

**Figure 15 sensors-23-04547-f015:**
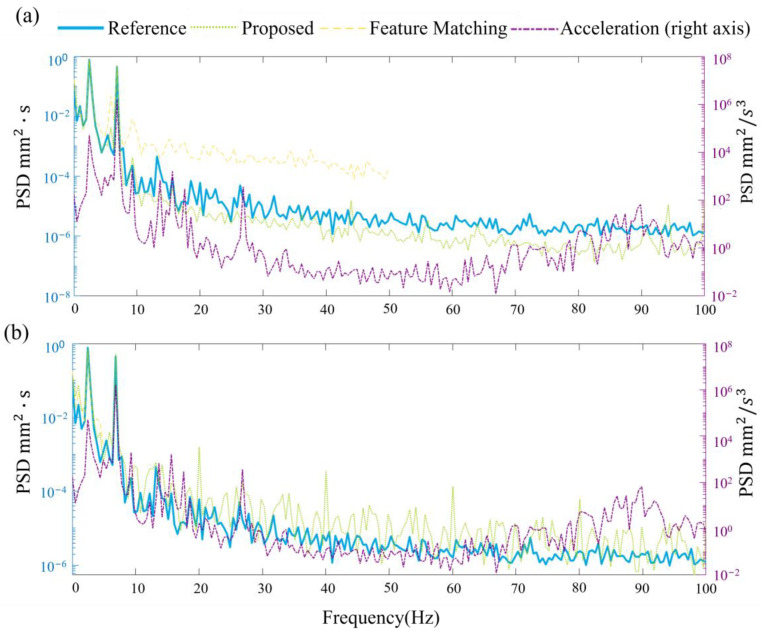
PSD for each measurement under earthquake excitation with different vision sample frequencies (**a**) 100 Hz (**b**) 10 Hz.

**Table 1 sensors-23-04547-t001:** Details of the cameras and sensors.

Type	Description
Camera	A Sony ILCE-7RM4 camera, featuring a resolution of 1920 × 1080 p, is utilized to capture the video of the structural vibration at a frame rate of 100 fps.
Laser displacement sensor (LDS)	A Panasonic HG-C 1200 micro laser distance sensor is employed to supply the ground-truth displacement data for the top floor, with a sampling rate of 500 Hz
Accelerometer	A KT-1100 accelerometer is employed to deliver the acceleration data for the top floor, with a sampling rate of 500 Hz.

**Table 2 sensors-23-04547-t002:** Compute time per frame of different algorithms.

Algorithm	Time Per Frame(s)
Proposed method	0.035
Feature matching method	0.172
KLT	0.009

## Data Availability

The data are available on request due to restrictions.

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
