# Peer review of "Vision and Vibration Data Fusion-Based Structural Dynamic Displacement Measurement with Test Validation"

_sensors, 2023, doi:10.3390/s23094547_

Round 1
Reviewer 1 Report
Review of the manuscript entitled “Vision and Vibration Data Fusion-based Structural Dynamic Displacement Measurement with Test Validation” by Cheng Xiu et. al.
Structural health monitoring is an important part of the overall chain of support for the construction in order to maintain its intended functions. Above many different methods applied to SHM, the ones monitoring the structure’s dynamic behaviour are particularly important as they deliver essential information on the structural condition. The measurement methods for dynamic testing may generally be divided into two categories: contact and non-contact ones. The latter ones are of particular importance as they are non-invasive and, in many cases, deliver the information from the entire field (so-called Full-Field optical techniques, such as Digital Image Correlation and others). Unfortunately, image-based techniques usually underperform from the perspective of sampling rate, primarily if high-resolution optical sensors are used. For that reason, the coupling of high-speed contact sensors and low-speed optical sensors is inevitable. In the presented paper, the authors propose the fusion of the contact type sensors – accelerometers with vision-based sensors for displacement monitoring. This combination results in achieving higher displacement measurement accuracy with an extended frequency range. From this perspective, the paper is worth publishing, and I recommend it. However, the text should be carefully improved, taking into consideration the following comments:
1) The authors propose the noise parameter q of a specific value in the Experimental Results section. I would expect some explanation, as to why it was this particular value. Does it come from their intuition, or experience, or was it estimated in some way?
2) In Figures 10, 13, 14a and 15, the authors show the plots with results obtained from different methods, and so I would expect that the colour of the lines in each of the plots for the same method remains the same, so it would be much easier for the reader to follow. E.g., in Fig. 10, the KLT method is indicated in red, whereas in Fig. 13, the red line belongs to the proposed method. This is misleading; Moreover, the bar plots comparing NRMSE (Fig. 12 and 14b) could also have the same colors as the lines in aforementioned plots;
3) The authors use the expression, “time history curves” in the text. What are they?
4) When the authors process visual data, they seek the natural targets in each figure. I understand that they process only ROI in every picture. Hence, is it possible to record only the ROI during the measurements? If so, the frequency of data capturing could be improved.
5) The authors discuss the similarities of PSD functions plotted in Fig. 15. Primarily, there are no colors indicating which line belongs to which method, which makes the plot hard to read. Assuming that the colors are the same as in Fig. 10, I agree that the proposed method is close to reference one in Fig. 15a. However, I would not be so sure about this in Fig.15b. Perhaps it would be a good idea to introduce some quantitative measure of two plots similarity. That would give objective proof of the superiority of the proposed method compared to others.
6) General text improvements: many grammatical, spelling and style errors may be found. Therefore it is recommended to proofread the manuscript with a native speaker. Some of the errors are as follows:
In the abstract, the first and third sentences are essentially the same;
Line 42: it should be vibrometers, not vibrometer;
Line 56: “import assumptions” phrase does not have sense;
Line 72: “can prevent” does not sound right in that context;
Line 78: it should be “Roberts”, not “Robert”;
Line 81: it should be “… several researchers have proposed …”;
Line 142: what are “nature targets”? Should it not be “natural”?
Line 144/145: should be “calibration”, not “calibrate”;
Line 148: The subchapter title should be “Calibration stage”;
Line 159 - 161: This paragraph has no sense in this place. Moreover, there is no Section 3.4 in the manuscript;
Line 176: should be “synchronizes”, not “synchronized”;
Line 336: should be “method”, not “mothed”;
6) General text improvements: many grammatical, spelling and style errors may be found. Therefore it is recommended to proofread the manuscript with a native speaker. Some of the errors are as follows:
In the abstract, the first and third sentences are essentially the same;
Line 42: it should be vibrometers, not vibrometer;
Line 56: “import assumptions” phrase does not have sense;
Line 72: “can prevent” does not sound right in that context;
Line 78: it should be “Roberts”, not “Robert”;
Line 81: it should be “… several researchers have proposed …”;
Line 142: what are “nature targets”? Should it not be “natural”?
Line 144/145: should be “calibration”, not “calibrate”;
Line 148: The subchapter title should be “Calibration stage”;
Line 159 - 161: This paragraph has no sense in this place. Moreover, there is no Section 3.4 in the manuscript;
Line 176: should be “synchronizes”, not “synchronized”;
Line 336: should be “method”, not “mothed”;
Reviewer 2 Report
This study proposes a data fusion method to combine both optical measurements and accelerometer readings for displacement estimation. The optical measurements employ the optical flow algorithm, while an acceleration-to-displacement conversion exploits the double integration. Then, the data fusion is realized by the asynchronous multi-rate Kalman filter. For verification, a scaled building is used to test the proposed method, which is also compared to the displacements obtained from a laser distance sensor, a feature-matching method, and the conventional KLT method. Finally, the results demonstrate the proposed method has better performance against the feature-matching and conventional LKT methods.
This manuscript is well organized. Before this manuscript is accepted for publication in this journal, the following comments should be carefully addressed.
Abstract
· The authors provide very good statements about the research gap; however, these statements seem too long.
· Moreover, the methodologies used in this study should also be included in the Abstract.
· The concluding remark can be more specific.
Introduction
· The third paragraph can be combined into the fourth one.
· The fifth paragraph can be integrated into the sixth one.
Section 2
· In Line 107, a window can be replaced by the region of interest (RoI).
· In Line 116, is Z defined correctly?
· In Line 127, what is ith frame?
· For the image pyramid, this method is used to decompose an image (i.e., without the idea of time). However, the statement in Lines 127-128 has nothing to do with the image pyramid. Therefore, please provide a more detailed procedure for the optical flow calculation. Moreover, Figure 2 is not really instructive, and please consider the other way to present the method.
Section 3
l In Section 3.1.1, the lens distortion model may need a reference.
l In Lines 160-161, using a double integral of accelerations to obtain displacements should be more specific because this calculation is irrelevant to optical measurements.
l The authors should provide a way to determine the band in the band-pass filter for the acceleration-to-displacement conversion.
l In Line 170, “the frequency” should be substituted into “the sampling frequency”. Moreover, is the resampling mentioned in Section 3.1.2 upsampling? If yes, please provide the reasons.
l In Section 3.1.3, are any superresolution methods involved?
l In Figure 5, the arrow lines connecting to the KLT block should be added with an adding block. Moreover, how is the predicted displacement (i.e., in Line 209) determined?
Section 4
l How is the noise parameter determined in Line 315?
l In Line 320, what do the authors mean by “linearly interpolating the data to 500 Hz”?
l How did the authors find “the forced vibration frequency…” in Lines 307-308?
l The authors can consider using BLWN excitation to calculate PSD or frequency response functions. Then, more findings can be obtained with respect to effective frequency ranges or error sources.
Some minor grammar errors are found. Please revise these errors accordingly.
Reviewer 3 Report
This paper presents a approach for measuring displacement by combining data from both vision-based and contact acceleration monitoring methods. The proposed method is validated through experiments designed by the authors, and the results demonstrate its superiority in accurately measuring displacement. The paper includes detailed descriptions of the method, equations, and figures, making it easy to understand and replicate the experiments. Overall, this paper is of value and deserves acceptance. However, before acceptance, the authors should address the following comments.
Major Comments:
1. In paragraph 4 of the Introduction section, the authors stated that the Kanade-Lucas-Tomasi method has two limitations. The reviewer has recommended adding a reference to support this claim. Additionally, the reviewer has suggested providing a brief explanation of the reasons behind these limitations. It would be helpful to address these suggestions to enhance the clarity and credibility of the paper.
2. In paragraph 5 of the Introduction section, the authors stated that the problems mentioned in the previous paragraph can be resolved by integrating accelerometer data. The reviewer has asked for clarification on whether this is a conclusion from prior research or a finding of the authors' work. If it is a result of previous research, the reviewer has suggested adding a reference to support this claim. If it is the main conclusion of this study, the reviewer has recommended placing this statement after the method introduction to provide better context. Otherwise, the statement may be difficult to comprehend.
3. The reviewer has inferred that the authors have proposed a new data fusion method. It would be beneficial to include a review of the literature to determine if there are any similar methods proposed by other researchers. If such methods exist, the authors should add a brief discussion of those studies. Furthermore, it is important to explain why fusing contact acceleration monitoring data and non-contact displacement recognition data is a significant method to study. The authors could provide a brief overview of the advantages of this approach over existing methods.
4. In Section 2, "A brief review of the Kanade-Lucas-Tomasi (KLT) method," the authors presented the theoretical framework of KLT. To help the reviewer understand the need for the proposed method, it would be beneficial to introduce the limitations of KLT in this section. The authors could provide one or two specific examples to illustrate the shortcomings of KLT. By highlighting the limitations of KLT, the authors can provide a more compelling case for developing a new method.
5. In section 3.3, the authors mentioned they will compare the results of the proposed method, a feature-matching-based method and the KLT tracker. The reviewer suggests adding an explanation of why the results from the last two methods are selected for this comparison. Are those two methods the best methods yet?
6. The authors presented a novel method that combines contact acceleration monitoring data and non-contact displacement recognition data, highlighting its advantages. However, the reviewer has raised a concern that the proposed method may also inherit the disadvantages of both techniques. For instance, the authors noted that the contact acceleration method is susceptible to device damage during dynamic testing. The authors should discuss potential strategies to mitigate these disadvantages or explain how the proposed method addresses these issues.
1. There are minor English errors. For example, line 228 “Among these three types…” lacks the subject. Please check the paper again for those small errors.
2. The font size of x and y axis and the legend of some figures are too small. Please make them larger.
Round 2
Reviewer 1 Report
As I stated in my review, I appreciate the topic discussed in the manuscript. I also appreciate the corrections made by the authors after the review. I recommend the paper for publication as it is. Just two minor corrections should be made:
Line 237: there is an error with the reference that cannot be found;
In Figure 15: in the line description, there is 'Vision' for red dashed line. Should not it be 'KLT'?
